# Mössbauer spectroscopy of a monolayer of single molecule magnets

Alberto Cini[1], Matteo Mannini [2], Federico Totti [2], Maria Fittipaldi[1], Gabriele Spina[1], Aleksandr Chumakov [3], Rudolf Rüffer[3], Andrea Cornia [4] & Roberta Sessoli [2]

The use of single molecule magnets (SMMs) as cornerstone elements in spintronics and quantum computing applications demands that magnetic bistability is retained when molecules are interfaced with solid conducting surfaces. Here, we employ synchrotron Mössbauer spectroscopy to investigate a monolayer of a tetrairon(III) ($Fe_4$) SMM chemically grafted on a gold substrate. At low temperature and zero magnetic field, we observe the magnetic pattern of the $Fe_4$ molecule, indicating slow spin fluctuations compared to the Mössbauer timescale. Significant structural deformations of the magnetic core, induced by the interaction with the substrate, as predicted by ab initio molecular dynamics, are also observed. However, the effects of the modifications occurring at the individual iron sites partially compensate each other, so that slow magnetic relaxation is retained on the surface. Interestingly, these deformations escaped detection by conventional synchrotron-based techniques, like X-ray magnetic circular dichroism, thus highlighting the power of synchrotron Mössbauer spectroscopy for the investigation of hybrid interfaces.

[1] Department of Physics and Astronomy and INSTM Research Unit, University of Florence, 50019 Sesto Fiorentino, Italy. [2] Department of Chemistry 'Ugo Schiff' and INSTM Research Unit, University of Florence, 50019 Sesto Fiorentino, Italy. [3] ESRF-The European Synchrotron, CS40220, 38043 Grenoble Cedex 9, France. [4] Department of Chemical and Geological Sciences and INSTM Research Unit, University of Modena and Reggio Emilia, 41125 Modena, Italy. Correspondence and requests for materials should be addressed to G.S. (email: gabriele.spina@unifi.it) or to A.C. (email: andrea.cornia@unimore.it) or to R.S. (email: roberta.sessoli@unifi.it)

Single molecule magnets (SMMs) are a very appealing class of nanomagnetic objects with potential application for spintronics[1–3] and quantum computing[4–6]. Their properties depend on the combination of a large molecular spin and an easy-axis magnetic anisotropy, which results in a double well energy potential, that opposes to the reversal of the magnetization[7]. Organization of these molecules on surfaces was the focus of considerable effort as a prerequisite to single molecule addressing[8–10]. In such studies, it was demonstrated that magnetic bistability persists and can even be enhanced on a surface[11, 12]. Especially efficient in boosting the memory effect of both molecules[13] and individual atoms[14] is deposition on thin insulating layers (e.g., MgO) rather than directly on the metal surface. Despite these remarkable results, the factors controlling magnetic bistability on surfaces remain still unclear. This is in part due to the limited number of experimental techniques that are sensitive enough to detect the magnetic properties of a monolayer (or less) of magnetic molecules. Most investigations rely on the use of X-ray absorption and magnetic circular dichroism (XMCD)[15] with synchrotron radiation, which has exceptional surface sensitivity and selectivity to the magnetism of the probed elements. In silico methods can also be of considerable aid in predicting the fine evolution of geometrical and magnetic structure of SMMs on a surface[16]. In such a framework, Mössbauer spectroscopy, beyond having an outstanding sensitivity to the coordination environment of the probed atom, is able to investigate the spin dynamics over timescales (1–1000 ns) much shorter than those accessible by XMCD. The technique was previously adopted to study the relaxation behavior of many SMM materials containing $^{57}$Fe as Mössbauer active nucleus[17, 18]. However, the standard use of radioactive sources limits the application domain of Mössbauer spectroscopy to bulk samples[19].

This limitation was overcome by the use of time-domain nuclear resonant scattering of synchrotron radiation at grazing angle: an example was the study of monolayers of metallic $^{57}$Fe grown on W(100) or embedded in layered systems[20–22]. However, with this technique complex spectra are expected for samples containing inequivalent iron sites and characterized by several electronic levels. Moreover, the radiation intensity customarily used in time-domain spectroscopy may damage molecular or biological compounds. Here, we show how energy-domain Mössbauer spectroscopy based on the high brilliance of synchrotron light can be used to probe the magnetism of a molecular monolayer, providing an unprecedentedly detailed picture of molecule-surface interaction effects on structural, electronic, and magnetic properties of the adsorbed SMMs.

Our study focused on the tetrairon(III) (Fe$_4$) SMMs, whose propeller-like molecular structure is sufficiently robust to withstand chemisorption on surfaces, processing by thermal sublimation[23], and the construction of single-molecule devices[24]. Indeed, Fe$_4$ complexes were the first SMMs to show magnetic hysteresis when deposited as a monolayer on a gold surface[11, 23]. They have a ground state with a total spin $S = 5$, while the first excited states are two $S = 4$ manifolds, which lie about 60 K higher in energy and are degenerate in case of perfect three-fold symmetry. A moderate easy-axis magnetic anisotropy, directed perpendicular to the plane of the four Fe$^{3+}$ ions, splits the ground spin manifold of about 15 K, which also corresponds to the height of the energy barrier for the reversal of the magnetization.

Here a monolayer of Fe$_4$ SMMs tethered to a polycrystalline gold substrate was investigated by synchrotron-based Mössbauer spectroscopy, which revealed magnetic features in zero field typical of SMM behavior. Thanks to the outstanding sensitivity of this spectroscopy to the coordination environment of the probed atom, we have also evidenced that molecules on surface undergo significant structural deformations, which are undetectable by

X-ray absorption techniques and scanning probe methods[25]. As predicted by ab initio molecular dynamics (AIMD) calculations[26], these modifications affect differently the four iron atoms. However, such local distortions partially compensate each other and the molecules retain on the surface an $S = 5$ ground state and a slow spin dynamics comparable to that of the bulk phase, thus justifying the magnetic robustness of this class of SMMs.

## Results

**Synchrotron Mössbauer Spectroscopy of Fe$_4$ SMMs.** In this work, we studied a Au-supported self-assembled monolayer of the same Fe$_4$ derivative (see Fig. 1) previously employed in XMCD investigations[27]. Its complete formula is [Fe$_4$(L)$_2$(dpm)$_6$], where Hdpm is dipivaloylmethane and H$_3$L is 7-(acetylthio)-2,2-bis(hydroxymethyl)heptan-1-ol, a thioacetyl-functionalized tripodal ligand that promotes chemisorption on gold substrates (see Fig. 1). The compound was characterized elsewhere[27] and was here prepared as a 95% $^{57}$Fe-enriched sample (see Methods section and Supplementary Methods). A monolayer deposit was obtained by chemisorption using a wet chemistry protocol (see Methods section) that guarantees the formation of a dense monolayer with no physisorbed material[28]. As a bulk-phase reference, a dropcast sample of the same compound, with an inhomogeneous thickness of about $100 \pm 50$ nm, was prepared.

The Synchrotron Mössbauer Source (SMS) available at ID18, the nuclear resonance beamline[29] of the European Synchrotron Radiation Facility (ESRF), was used to record Mössbauer spectra of both dropcast and monolayer Fe$_4$ samples. The characteristic device of the SMS is an iron borate ($^{57}$FeBO$_3$) crystal, kept at a temperature close to its Néel temperature (see Fig. 1d). By means of a pure nuclear reflection by the crystal, the synchrotron light coming out from the monochromators is filtered into an $^{57}$Fe-resonant narrow single line[30, 31]. In contrast to common radioactive sources, the radiation generated by the SMS is a needle-like collimated beam with small (~mm) size, which can be further focused to spot sizes of micrometric lateral dimensions[31]. This allows working in grazing incidence geometry to achieve the necessary sensitivity to investigate ultra-thin films. Furthermore, the radiation is fully polarized and recoilless, and sufficiently high photon fluxes can be obtained (see Supplementary Methods for details).

Mössbauer spectra were recorded by collecting the scattered radiation resulting from the reflection on the sample surface at a grazing incidence (see Fig. 1d). Due to the structure of our samples (low-$Z$ film on a high-$Z$ substrate, $Z$ being the atomic number), the grazing incidence reflection occurs from the surface of the substrate, and the molecular layer produces absorption lines. In the case of small grazing angles, the spectra can be treated as those obtained in a standard setup in transmission geometry, provided that the effective thickness of the samples is multiplied by a factor $2/(\sin\theta)$, where $\theta$ is the grazing angle between the surface of the sample and the direction of the incoming radiation. In our experimental conditions, working at $\theta \sim 0.1°$ provides a 1100-fold amplification factor, which is essential for studying molecular monolayers by SMS.

Mössbauer spectra of the dropcast and monolayer samples were recorded in the temperature ranges 2.2–40 and 2.2–11 K, respectively, and are shown in Fig. 2. At the lowest explored temperature, the spectrum of the dropcast sample (Fig. 2a) exhibits a six-line pattern, as expected for a Fe-containing sample experiencing slow spin fluctuations compared to the Mössbauer timescale. However, the lines towards the extremes of the spectrum are further split, indicating that inequivalent iron sites are present in the structure; this is in line with previous reports on

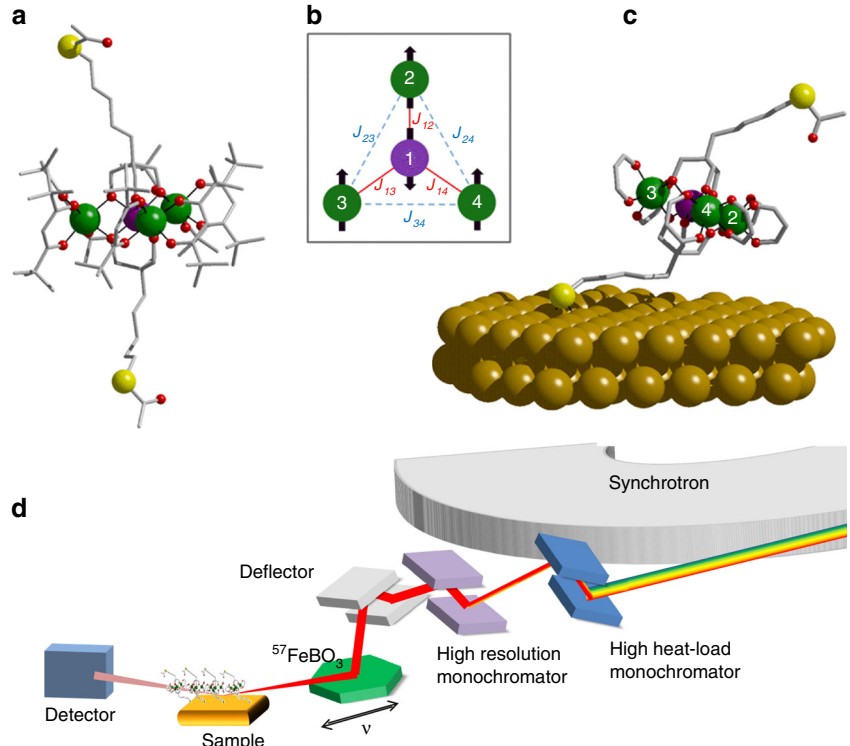

**Fig. 1** Structure of the investigated systems and experimental setup. **a** Structure of $Fe_4$ SMM in the crystalline phase (color code: Fe atoms in green and violet, O in red, S in yellow, C as pale gray sticks, H atoms omitted for clarity); **b** scheme of the magnetic core, where the dominating antiferromagnetic interactions (in red) lead to an $S = 5$ ground state; **c** view of the structure of the $Fe_4$ SMM tethered to Au(111) through the deprotected thioacetyl termination, as obtained by ab initio molecular dynamics calculations[26]. Hydrogen atoms and *tert*-butyl groups on dpm⁻ ligands have been omitted for clarity; **d** scheme of the experimental setup of the Synchrotron Mössbauer Source at the ID18 beamline of ESRF

standard Mössbauer characterization of unfunctionalized $Fe_4$ clusters in the bulk phase[18].

With increasing temperature, distortions of the spectrum become evident. Increased thermal fluctuations and the population of excited spin levels cause a progressive collapse of the spectrum into a single central structure at 40 K.

The monolayer's spectra are qualitatively similar, though about two orders of magnitude less intense (Fig. 2b), in agreement with the reduced number of $^{57}Fe_4$ molecules probed by the radiation, (see Supplementary Figs.1–4 for a direct comparison between selected spectra of the two samples).

**Fitting of the Mössbauer spectra**. Given that the $Fe_4$ molecule has total spin state $S = 5$, we expect each iron atom to contribute to the spectra with six hyperfine-split lines per $|\pm M_S\rangle$ doublet ($M_S = 1, 2, \ldots 5$) plus two quadrupole-split lines arising from the $|M_S = 0\rangle$ state, thus yielding $4 (6 \times 5 + 2) = 128$ lines. Some of these transitions may coincide because of the molecular symmetry. In particular, while the central ion (Fe1) experiences a quite distinct coordination environment, the other iron atoms reside in approximately the same coordination environment. Despite this idealized three-fold symmetry, $Fe_4$ derivatives in the solid state often possess a binary symmetry axis lying along two iron atoms. Therefore, for a quantitative analysis of the dropcast sample spectra we considered that the Mössbauer absorption cross-section is given by the superposition of the contributions coming from three inequivalent iron ions, with intensity ratio 1:1:2. Each contribution was evaluated as described in earlier Mössbauer bulk-phase experiments[17, 18] and characterized by a number of parameters (see Methods and Supplementary Note 1). These

include hyperfine interactions for each iron site (the isomer shift with respect to α-Fe, the quadrupole shift, and the hyperfine fields due to the iron electronic spin), magnetic anisotropy and magnetic exchange interactions governing the transition rates between different spin sublevels of the system, as well as sample-specific parameters, like thickness and texture. The latter is a parameter quantifying a possible macroscopic orientation of the molecules on the surface. In particular, the possibility to sense a texture, which has a signature in the relative intensities of the lines, is based on the orientation of the quantization axis of the molecule with respect to the γ-ray polarization.

For the dropcast sample, the agreement between the experimental and calculated data (Fig. 2a) is reasonably good in the whole temperature range. In Fig. 3a, the cross-section contributions of each $Fe^{3+}$ site to the spectrum taken at 2.2 K are shown, with best fit parameters reported in Table 1 (the values obtained from the fits of higher temperature spectra are reported in Supplementary Tables 1–3, while residuals are shown in Supplementary Figs. 5 and 6).

As expected, the Mössbauer parameters associated to three out of the four $Fe^{3+}$ ions are very similar to each other and their contributions overlap extensively, in agreement with an almost ideal three-fold symmetry of the cluster. The contribution with a significantly different quadrupole interaction and a smaller hyperfine field can be thus safely assigned to the central $Fe^{3+}$ ion on the basis of its different coordination environment.

In general, the hyperfine field values are proportional to the number of unpaired 3d electrons for each iron ion ($i = 1–4$) and thus to the spin projection along the local direction of the quantization axis. However, exchange interactions introduce spin fluctuations at frequencies higher than the top limit of the

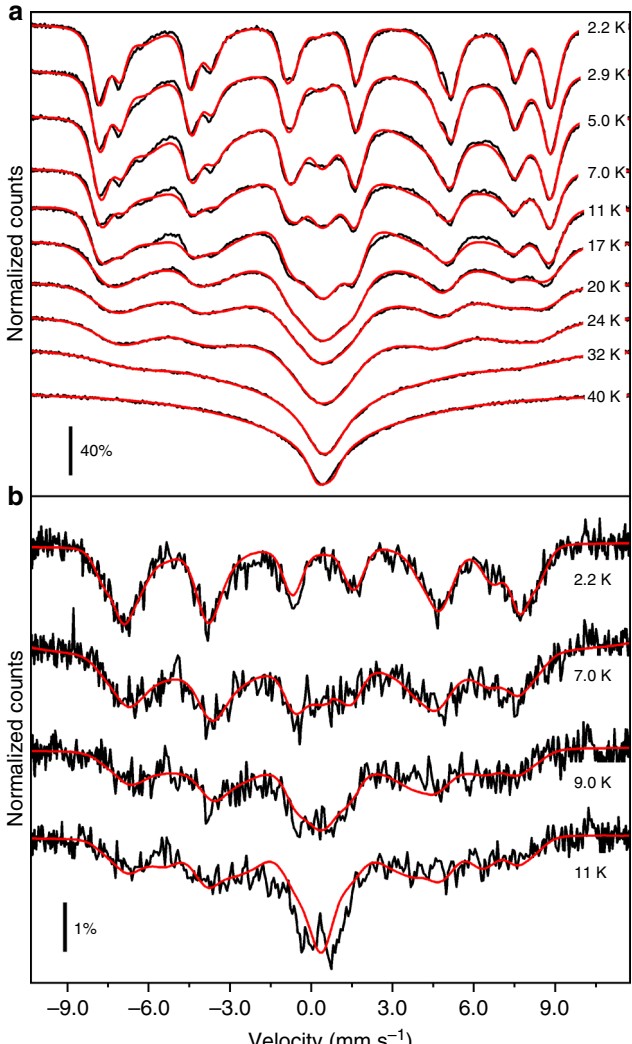

**Fig. 2** Mössbauer spectra of the dropcast and monolayer samples for various temperatures. **a** Experimental spectra (black lines) of the $Fe_4$ dropcast sample and best fit curves (red lines). **b** Experimental spectra (black lines) of the $Fe_4$ monolayer sample and best fit curves (red lines). The velocity axis values are relative to the α-Fe standard

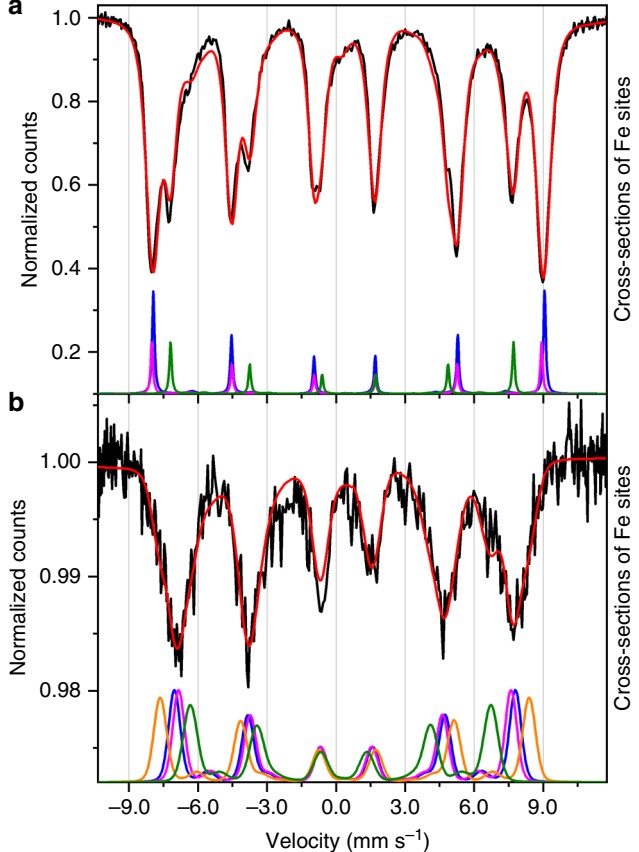

**Fig. 3** Deconvolution of Mössbauer spectra in individual $Fe^{3+}$ contributions. **a** Mössbauer spectrum at 2.2 K of the dropcast sample (black line) and best fit curve (red line); the cross-sections of the three inequivalent $Fe^{3+}$ sites expected in a twofold-symmetric molecule are shown in green, magenta, and blue and were calculated with the parameters listed in Table 1 (columns from left to right, respectively). **b** Mössbauer spectrum at 2.2 K of the monolayer sample (black line) and best fit curve (red line); the cross-sections of the four inequivalent $Fe^{3+}$ sites (green, orange and blue/magenta) were calculated with the parameters listed in Table 1 (columns from left to right, respectively)

Mössbauer time window, so that the measured hyperfine fields are proportional to the time average of the spin projections, $\langle s_z^i \rangle$. The latter are affected by both magnetic exchange interactions and magnetic anisotropies, as most simply described by the spin Hamiltonian:

$$ H = \sum_{i,j=1,4}^{i<j} J_{ij}\mathbf{s}_i \cdot \mathbf{s}_j + \sum_{i,j=1,4}^{i<j} \mathbf{s}_i \cdot \overline{d_{ij}} \cdot \mathbf{s}_j + \sum_{i=1,4} \mathbf{s}_i \cdot \overline{D_i} \cdot \mathbf{s}_i \tag{1} $$

The first term represents the isotropic exchange interaction, the second one contains the anisotropic part of intramolecular interactions, which are mainly of dipolar origin in assemblies of high spin $3d^5$ ions, while the third one takes into account single ion magnetic anisotropy. Nearest-neighbour antiferromagnetic exchange interactions between the central ($i = 1$) and peripheral ($i = 2, 3,$ and 4) ions provide the leading term responsible for the ground $S = 5$ molecular state. This state features a ferrimagnetic-like arrangement of the spins, with the central and peripheral spins pointing in opposite directions (Fig. 1b). Because of

magnetic anisotropy, the total spin vector in the ground state lies preferably collinear to the idealized three-fold molecular axis and the $S = 5$ multiplet is thus split approximately according to the giant-spin Hamiltonian[7]:

$$ H_{S=5} = D\left[S_z^2 - \frac{1}{3}S(S+1)\right] + E\left(S_x^2 - S_y^2\right) \tag{2} $$

where $D < 0$ and $E$ are the second-order axial and transverse magnetic anisotropy parameters, respectively. Assuming three-fold symmetry, i.e. $E = 0$, the spin projection values within the ground $M_S = \pm 5$ doublet are $\langle s_z^1 \rangle = \mp 2.0833$ and $\langle s_z^2 \rangle = \langle s_z^3 \rangle = \langle s_z^4 \rangle = \pm 2.3611$ (in $\hbar$ units). The experimental hyperfine fields for the central and peripheral ions in the dropcast sample (Table 1) are in good agreement with the so-called $22\langle s_z^i \rangle$ rule for the Fermi contact field[32] (Supplementary Fig. 7). This indicates that the Mössbauer spectrum at 2.2 K probes intact $Fe_4$ complexes in their $M_S = \pm 5$ ground doublet.

Moreover, below 40 K, the thermal evolution of the spectra (Fig. 2a) likely reflects the contribution of excited states within the $S = 5$ manifold as determined by two main, and somehow distinguishable, mechanisms. Indeed, on increasing temperature,

**Table 1 Mössbauer parameters extracted from the fitting of the spectra at $T = 2.2$ K and from ab initio calculations**

| | Parameter | Central site | Peripheral sites[f] | | |
|---|---|---|---|---|---|
| **Dropcast** | Isomer shift (mm s$^{-1}$)[a] | 0.409 (2) | 0.418 (4) | 0.467 (2) | 0.467 (2) |
| | Electric quadrupole shift (mm s$^{-1}$)[b] | −0.149 (2) | 0.045 (4) | 0.094 (2) | 0.094 (2) |
| | Hyperfine magnetic field (T) | 46.39 (1) | 52.72 (2) | 52.92 (2) | 52.92 (2) |
| **Dropcast (theory)** | | **Fe1** | **Fe2** | **Fe3** | **Fe4** |
| | Hyperfine magnetic field, $A_{iso}$ (T)[c] | 46.90 | 52.34 | 54.49 | 53.32 |
| **Monolayer** | Isomer shift (mm s$^{-1}$)[a] | 0.27 (5) | 0.43 (4) | 0.42 (6) | 0.42 (6) |
| | Electric quadrupole shift (mm s$^{-1}$)[b] | −0.07 (4) | −0.06 (2) | −0.03 (2) | −0.03 (2) |
| | $\sigma$ of Gaussian broadening (mm s$^{-1}$) | 0.28 (6) | 0.25 (5) | 0.2 (1) | 0.2 (1) |
| | Hyperfine magnetic field (T) | 40.7 (4) | 49.9 (3) | 45.6 (9) | 45.6 (9) |
| **Monolayer (theory)** | | **Fe1** | **Fe2** | **Fe3** | **Fe4** |
| | Hyperfine magnetic field, $A_{iso}$ (T)[c,d] | 43.68 (W1) | 52.11 (W1) | 52.22 (W1) | 53.04 (W1) |
| | | 45.09 (W3) | 52.50 (W3) | 51.73 (W3) | 49.04 (W3) |
| | $\Delta A_{iso}$ (T) due to Au substrate[e] | −1.77 | −0.28 | −0.55 | −0.60 |

[a]With respect to α-Fe
[b]The formula of the electric quadrupole shift is $\epsilon = \frac{e^2 qQ}{4}\left(\frac{3\cos^2\theta - 1}{2}\right)$, where $\theta$ is the angle between the electric field gradient principal axis and the direction of the hyperfine magnetic field at the iron atom
[c]The calculated hyperfine fields (Fermi contact terms) take into account the $\langle s_z^i \rangle$ value and were scaled by a factor of 1.81 (see text and Methods section)
[d]Geometries resulting from two different AIMD trajectories as in[26], i.e., walker 1 (W1) and walker 3 (W3), were considered
[e]Reduction of $A_{iso}$ (T) values due to pure electronic effects induced by the substrate. The $\Delta A_{iso}$ values were calculated as $22\Delta\langle s_z^i \rangle$ (averaged over all AIMD walkers) from the variation of the spin density, computed at the GPW-DFT level, between-on-surface and extrapolated geometries
[f]The numbering of peripheral sites only applies to computed values; experimental parameters in the last column apply to two indistinguishable sites, errors are given in parentheses

the thermal population of states with $|M_S| < 5$ increases the number of thermally accessible transitions with a corresponding change in the overall intensity of the lines. On the contrary, the increased transition rate between different states causes changes in shape and shifts the lines towards the center of the spectrum, as illustrated in Supplementary Note 2. Inspection of Fig. 2a shows that the thermal evolution of the spectra is dominated by the population of excited spin states below 11 K (slow relaxation regime), and by the increased transition rates above 11 K (intermediate relaxation regime). The latter is evident from the collapse of the spectrum towards a central line superimposed to a V-shaped baseline (see Supplementary Note 2 and Supplementary Fig. 8 in which the two different behaviors are simulated at 11 K). As a consequence of this, the physical parameters describing both effects could be extracted from the fit of the spectra. The best fit values, $D = -0.605(7)$ K and $E = 0.003(1)$ K, well agree with those resulting from magnetic data for the bulk crystalline phase[27].

The transition rates among the spin states of the $S = 5$ manifold are determined both by the $E$ term, promoting the tunnelling between the two potential wells, and by the interaction of the molecular spin with the thermal bath. To take this last interaction into account, a spin–bath interaction term $H_{sb}$ has to be included in Eq. 2:

$$H_{sb} = F_b Q_S(S_x, S_y, S_z) \qquad (3)$$

$H_{sb}$ is expressed as the product of two operators ($Q_S$ and $F_b$), which act on the molecular spin and lattice states, respectively. More details on these two operators can be found in Supplementary Notes 3 and 4. Considering that the matrix elements of $Q_S$ between two given spin states do not depend on temperature, the thermal dependence of the matrix elements of $H_{sb}$ is linked only to the thermal dependence of the matrix elements of $F_b$ and thus to the physical features of the thermal bath and to the specific interaction scheme[17]. In particular, the parameters determining the thermal dependence of the spectra are proportional to the Fourier transform of the correlation function of $F_b$ evaluated at the transition frequencies. This quantity can be evaluated at zero frequency in the usual approximation where the linewidth of bath states is larger than the energy separation between the electronic

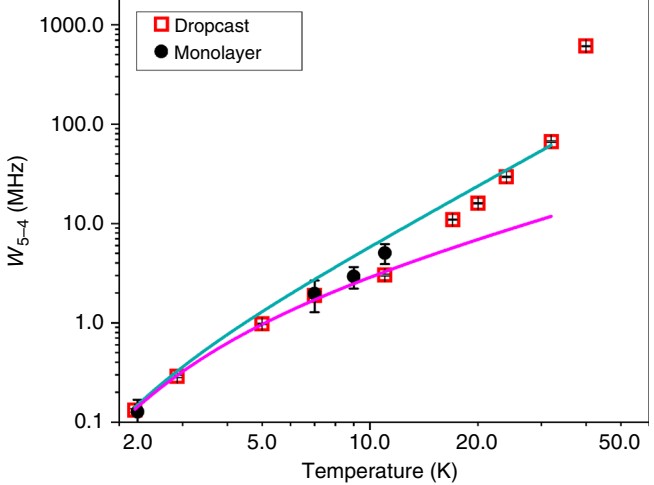

**Fig. 4** Spin dynamics in the dropcast and monolayer samples. Transition rate $W_{5-4}$ between the states $M_S = \pm 5$ and $M_S = \pm 4$ extracted from the fits of the Mössbauer spectra of the dropcast sample (red squares) and of the monolayer sample (black dots) as a function of temperature. The magenta line represents the fit of the data for the dropcast sample at $T \leq 11$ K assuming a direct process as in Eq. (4), while the cyan line corresponds to the more general function reported in Eq. (5)

spin states. More details are provided in Supplementary Note 4, where both direct (single-phonon) and indirect (multi-phonon) processes are considered.

In the approximation introduced above the spin dynamics depends on the transition rates between the electronic spin states of the molecule. To analyse the spin dynamics as a function of temperature, the transition rate $W_{5-4}$ between $M_S = \pm 5$ and $M_S = \pm 4$ states was chosen as a fit parameter. All other transition rates between two generic $i$ and $j$ states were derived according to the relations $W_{j-i} = W_{5-4}|\langle i|Q_S|j\rangle|^2/|\langle 5|Q_S|4\rangle|^2$ and $W_{i-j} = W_{j-i}\exp((E_i - E_j)/(k_B T))$.

In Fig. 4, we present the temperature dependence of $W_{5-4}$, together with a fit of the data up to 11 K with a function of the form:

$$W_{5-4} = A/(\exp(B/k_B T) - 1), \qquad (4)$$

which is representative of a direct process. This analysis of low temperature transition rates yields $B = 6.8(4)$ K and $A = 2.8(3) \times 10^6$ Hz. For temperatures >11 K, the increase of $W_{5-4}$ is more rapid than predicted by Eq. (4) suggesting two-phonons processes. These can be taken into account using the expression:

$$W_{5-4} = A \exp(B/k_B T)/(\exp(B/k_B T) - 1)^2 \qquad (5)$$

that describes a Raman process due to optical modes. Equation (5) coincides with Eq. (4) for $T \ll B$, while for $T > B$ it gives a $T^2$ dependence (see Supplementary Note 4)[19]. The fit, shown in Fig. 4, was performed keeping the $B$ parameter fixed at the value extracted from low temperature data. Equation (5) is qualitatively in agreement with the experimental data in a wide range of temperatures, suggesting that the spin dynamics is related to an optical mode with an energy of about 7 K (or, equivalently, a frequency of about 0.1 THz). A rapid increase of the transition rate is observed at the highest investigated temperature (40 K) where other total spins states start to be significantly populated.

Passing to the monolayer sample, the analysis of the spectra has been performed following the same strategy used for the dropcast sample. From the fitting of the monolayer spectrum taken at the lowest temperature, a surface density of ~3 $^{57}$Fe ions per nm$^2$ was estimated. This value agrees, within the experimental uncertainty of the technique, with the one evaluated by an STM study of Fe$_4$ molecules evaporated on a gold single crystal (~2 $^{57}$Fe ions per nm$^2$)[23]. Therefore, our study represents the first successful investigation of a monolayer of molecules by means of Mössbauer spectroscopy.

The best fit calculated curves are shown in Fig. 2b for each sampled temperature, while residuals are presented in Supplementary Figs. 9 and 10. The individual Fe$^{3+}$ cross-section contributions to the spectrum taken at 2.2 K are shown in Fig. 3b, with best fit parameters reported in Table 1. To comply with a possible reduction of symmetry of the cluster four non-equivalent Fe sites were considered. However, two contributions are identical within error (blue and magenta lines in Fig. 3b) and only three non-equivalent Fe sites are thus reported in Table 1. The parameters in Table 1 were kept fixed in the analysis of the spectra acquired at higher temperatures, while the transition rate ($W_{5-4}$) and the $D$ parameter were let free to vary.

The texture value extracted from each spectrum revealed no preferential orientation of the molecules on the substrate. In fact, imposing the same oriented grafting as detected by X-ray absorption experiments on Au(111)[27] fails to satisfactorily reproduce the data (see Supplementary Note 5 and Supplementary Fig. 11). This discrepancy is however not unexpected, since a reconstructed, atomically flat Au(111) surface, known to promote the preferential orientation detected by X-ray natural linear dichroism[27], could not be used for the Mössbauer experiments (see Methods section).

As a distinctive feature of the monolayer's spectra, all the lines of each sextet are characterized by approximately the same linewidth, although larger than found in the dropcast sample. Two mechanisms can be responsible for the observed line broadening. Faster transition rates between spin levels broaden the lines, but, at the same time, promote a collapse towards the center of the spectrum, which is not experimentally observed in the 2.2 K spectrum. Moreover, to reproduce the temperature dependence of the spectra an unphysical decrease of transition rates upon increasing temperature would be required. An alternative source of broadening is the presence of an inhomogeneous distribution of electric parameters (quadrupole shift and/or isomer shift). A significant distribution in hyperfine fields can be safely ruled out as it would affect the external lines approximatively six times more than the internal ones (see Supplementary Note 2 for details).

The presence of inhomogeneous broadening in the electric parameters was thus taken into account by assuming Gaussian lines with standard deviation of the order of 0.2–0.3 mm s$^{-1}$, as estimated from the lowest temperature spectrum, which is less affected by spin dynamics. If the observed broadening is ascribed to a distribution of the ligand contributions to the quadrupole interaction, the ligand donor atoms must undergo displacements of the order of 0.04 Å from their average positions. Furthermore, all the magnetic hyperfine field values are significantly smaller than those found in the dropcast sample.

As a final point, the spin dynamics of the grafted Fe$_4$ molecules was addressed by investigating the temperature evolution of the spectra (see Fig. 2b). Despite the much worse signal-to-noise ratio, it is evident that, on increasing the temperature, the external lines lose intensity faster than in the dropcast sample. This suggests that the |$D$| value is reduced upon grafting, as directly confirmed by fitting the Mössbauer spectrum recorded at the lowest temperature ($D = -0.49(6)$ K, with the $E/D$ ratio held fixed at the value found in the dropcast sample to avoid over-parametrization in the fitting of poorly resolved spectra). At the same time the transition rate $W_{5-4}$ extracted from the fit of the spectra (see Fig. 4) is comparable with that found for the dropcast sample. Thus, Mössbauer data of the monolayer sample indicate that the deposition on a surface does not significantly modify the spin dynamics, as suggested by previous XMCD evidences[27].

**Ab initio calculations**. Qualitative and quantitative analyses of the recorded Mössbauer spectra point to significant alterations induced by the chemical grafting on the surface. AIMD calculations, recently performed by some of us on the same molecule grafted on the Au(111) surface[26], can provide a valuable insight into the observed differences. Indeed, the picture emerging from the AIMD investigation was that van der Waals interactions between the gold substrate and the aliphatic tether of the tripodal ligand cause significant distortions of the magnetic core. Such interactions induce several accessible energy minima in the potential energy surface of the system. The predicted scenario closely matches that emerging from the Mössbauer spectra of the monolayer sample. Remarkably, the standard deviation of Fe–O bond lengths, estimated over eight different AIMD trajectories or walkers (see Supplementary Table 4), has the order of magnitude ($10^{-2}$ Å) required to account for the observed line broadening through a distribution of quadrupole shifts.

A pronounced variation of Fe–O bond lengths among the different walkers is observed for the central site (Fe1) and for the Fe atom closer to the aliphatic chain tethered to the gold substrate, namely Fe3 in Fig. 1c. Moreover, AIMD calculations showed that both single ion magnetic anisotropies and exchange interactions are affected by the grafting process (see Supplementary Table 5 and ref. [26]). However, these modifications partially compensate each other so that the molecule retains an $S = 5$ ground state and an easy axis magnetic anisotropy.

Focusing now on the hyperfine field experienced by the $^{57}$Fe nuclei, the process of chemical grafting on the gold surface can affect both the individual $\langle s_z^i \rangle$ values, by altering intramolecular exchange interactions in Eq. (1), and the extent of electron delocalization on the nuclei. For the eight investigated AIMD trajectories[26] the local $z$ components in the $M_S = \pm 5$ doublet are on average smaller in absolute value than in the bulk phase (Supplementary Fig. 7). The difference is more clearly

visible on the central site Fe1 (2.087 vs. 2.137 $\hbar$) than on peripheral sites (2.361 vs. 2.378 $\hbar$). Furthermore, Fe3 shows the largest spread of $\langle s_z^i \rangle$ values ($\pm 5\%$) over the eight walkers. In order to assign the observed hyperfine fields to specific $Fe^{3+}$ ions, we assumed that the hyperfine field is an increasing function of $|\langle s_z^i \rangle|$ averaged over the eight walkers. A linear trend was obtained (Supplementary Fig. 7), the slope of which is similar to that of the bulk sample, further indicating that $Fe_4$ complexes retain their magnetic features on surface. However, all values seem to be shifted on average by 6 T towards smaller fields.

To shed some light on this aspect, further ab initio calculations of the isotropic hyperfine magnetic fields were here performed. We considered the optimized geometry in the crystalline phase and two of the geometries obtained by AIMD trajectories after extrapolating the $Fe_4$ complex from the underlying gold surface. The computed hyperfine magnetic fields, reported in Table 1, are dominated by the Fermi contact term. The time-demanding relativistic spin orbit coupling (SOC) contributions were not computed as they are expected to be negligible for high spin $Fe^{3+}$. In agreement with Mössbauer spectra, the lowest hyperfine magnetic field is computed for the central iron in both bulk-phase and on-surface molecules (see Table 1), primarily as a consequence of the different $\langle s_z^i \rangle$. The hyperfine fields are on average reduced upon grafting. The strongest reduction (−2.5 T) is calculated for the central iron atom, but is much smaller than found experimentally.

The above described ab initio calculations only include surface-induced structural deformations, but neglect electronic effects of the metal surface. Their inclusion is not possible with the same accuracy, therefore the local spin densities were computed by pseudo-potential Gaussian Plane Waves (GPW-DFT) calculations by considering both the $Fe_4$ molecule and 314 gold atoms of the underlying slab.

To avoid spurious effect due to the use of pseudopotentials, these spin densities were compared to the ones obtained, at the same level of calculation, on molecules with the same geometry, but removing the gold slab—also referred to as extrapolated geometry. The results, averaged over the eight investigated AIMD trajectories, are reported in Supplementary Table 6 (for the individual values see Supplementary Table 7). They indicate that a small, but sizable reduction of the spin density occurs due to a purely electronic effect of the gold substrate, thereby hinting to a decreased polarization of core $s$ orbitals of iron ions. The expected reduction of hyperfine fields, reported in Table 1, is of the order of 2 T for Fe1 and significantly smaller for the remaining ions. Though a similar investigation considering all-electron basis is not affordable, the observed reduction of hyperfine magnetic fields on the surface can be ascribed to both geometrical and electronic perturbations induced by the grafting process.

## Discussion

Synchrotron Mössbauer spectroscopy applied here to a monolayer of molecules has provided unprecedented insights in the process of chemical absorption of a complex molecule on a surface. This has been achieved thanks to the largely increased sensitivity—without significant loss in spectral resolution—that characterizes the SMS as compared with a conventional Mössbauer setup. The resulting scenario is that significant deformations of the magnetic core are induced by the interaction with the substrate, even in the relatively well-protected and rigid $Fe_4$ structure. These deformations emerge clearly from state-of-the-art ab initio molecular dynamics modelling, but are undetectable by conventional synchrotron based magneto-optical techniques, such as XMCD, as well as by more local probes, like scanning probe techniques. In the latter case, in fact, the tip of the microscope mechanically induces deformation, thus altering the molecular structure[25]. In surface-supported $Fe_4$ complexes, SMM behavior is retained with comparable magnetization dynamics to the bulk phase. This observation can be reconciled with the evidences of surface-induced structural modifications because local distortions at different iron sites are calculated to partially compensate each other. This makes the $Fe_4$ class of molecules particularly well suited for organization on surface and insertion in single molecule devices. In this sense, synchrotron Mössbauer spectroscopy could be employed in the investigation of other types of molecules assembled in monolayer architectures in view of the rational design of hybrid interfaces[33] and single molecule spintronic devices.

## Methods

**Sample preparation**. The synthesis of [$^{57}Fe_4(L)_2(dpm)_6$] was based on the previously reported[27] preparation of [$Fe_4(L)_2(dpm)_6$] and is described in detail in Supplementary Methods, while the mass spectra of $^{57}Fe$ enriched vs. natural samples are reported in Supplementary Fig. 12. For the preparation of the monolayer sample a 20 nm thick polycrystalline gold film evaporated on silicon with a 5 nm Ti decoupling layer (SSENS, Enschede, NL) was immersed in a 2 mM solution of [$^{57}Fe_4(L)_2(dpm)_6$] in dichloromethane and incubated under controlled atmosphere for 20 h. This type of substrate, selected for its flatness and rigidity, is not compatible with hydrogen flame annealing, used to promote the surface reconstruction of Au(111) and the consequent preferential orientation of the $Fe_4$ molecules in the monolayer[27]. After incubation the slide was washed with pure dichloromethane in order to remove any physisorbed overlayers, dried under argon atmosphere and directly mounted on the SMS sample holder. No XAS/XMCD data are available for a monolayer assembled on the substrates employed here. However, we recall that $Fe_4$ molecules chemisorbed on well reconstructed Au(111) surfaces or tethered to Au nanoparticles[34] both give XAS/XMCD spectra that do not differ appreciably from those recorded in the bulk phase.

The dropcast film was prepared starting from the same kind of substrate and a freshly prepared 2 mM solution of [$^{57}Fe_4(L)_2(dpm)_6$].

**Mössbauer experiments**. Mössbauer spectra were measured at ID18, the nuclear resonance beamline[29] of the European Synchrotron radiation facility (ESRF), taking advantage of the Synchrotron Mössbauer Source (SMS)[30, 31]. For the present study an acceptable compromise between linewidth and acquisition time was achieved by setting the linewidth at half maximum (FWHM) at a value approximately three times larger than for a radioactive source. In these conditions, an intensity of about $1.5 \times 10^4$ photons per second was obtained on the 14.4 keV γ-line. The FWHM was evaluated by taking a Mössbauer spectrum of a single-line absorber ($K_2Mg^{57}Fe(CN)_6$) of known thickness before and after each Mössbauer measurement of the samples. Moreover, the isomer shift of the SMS line, as measured from the single-line spectra, was of 0.709 mm s$^{-1}$ with respect to conventional α-Fe. Mössbauer spectra were recorded by collecting the radiation reflected by the sample surface in a grazing incidence geometry. For both samples an incidence angle θ ~ 0.1° (with a spot size of ca 18 μm in both dimensions) was chosen, after measuring the reflectivity of the samples as a function of θ. Mössbauer spectra were measured at different temperatures in the range 2.2–40 K and 2.2–11 K for the dropcast and the monolayer samples, respectively, using the superconducting He-exchange gas cryo-magnetic system. Mössbauer spectra were recorded without applying any external magnetic field. The contribution of Fe impurities in beryllium collimating lenses of the beamline, as detected in an empty-can Mössbauer spectrum, i.e., with no mounted sample (see Supplementary Fig. 13), was subtracted from the experimental spectra.

**Fitting method**. The fitting of Mössbauer spectra required the determination of the absorption cross-section of the samples (further details are provided in Supplementary Note 1). In general, the total cross-section is the superposition of a number of contributions equal to the number of inequivalent Fe sites; however, symmetry considerations can reduce the number of distinct contributions and consequently change their relative intensities; for example, for the $Fe_4$ dropcast sample three contributions were used, with relative intensities in the ratio 1:1:2. The hyperfine Hamiltonians acting on the inequivalent iron nuclear sites are characterized by different static components (i.e., magnetic fields, electric quadrupole tensors, and isomer shifts), but share the same dynamical parameters, which reflect the interaction of the molecular total spin states with the thermal bath.

When the molecular spin undergoes transitions between its states, the spins at the individual iron ions change their $z$-components simultaneously; consequently, the iron nuclei are subjected to stochastic changes in the hyperfine field magnitude, though the ratios between the hyperfine fields at the different iron sites remain unchanged. To evaluate the various contributions to the Mössbauer cross-section, Liouville super-operators associated with the spin and the spin-thermal bath Hamiltonians acting on all the possible transitions between the hyperfine states,

were introduced (further details are provided in Supplementary Notes 2 and 3). The procedure required the evaluation of the eigenvectors and left and right eigenvalues of a set of non-Hermitian matrices of order 88. Moreover, broadenings of the cross-section contributions were introduced to describe inhomogeneous Gaussian distributions of hyperfine electric parameters.

The resulting total cross-section as a function of the energy expressed in mm s$^{-1}$ ($\sigma(\omega)$) was then inserted into the transmission integral function in order to take into account the dependence of the Mössbauer spectra on the sample effective thickness. The complete expression used to fit the spectra was

$$Y(\nu) = N_b(\nu)\left\{1 - \int_{-\infty}^{\infty} L_2^S(\omega - \nu, S)\left[1 - \exp(-t_a^{SMS}\sigma(\omega))\right]d\omega\right\} \qquad (6)$$

where $\nu$ and $N_b(\nu)$ are the transducer velocity and the spectrum baseline, respectively[35]. Moreover $L_2^S(\omega - \nu, S)$ is the squared Lorentzian distribution[31], centered at $\nu$ and having $\Gamma_S$ as FWHM, used to describe the source line shape. Finally, $t_a^{SMS}$ is the effective thickness of the sample, obtained multiplying the sample effective thickness in a conventional Mössbauer setup by the factor $2/(\sin\theta)$, where $\theta$ is the grazing angle between the surface of the sample and the direction of the incoming radiation. The correctness of the procedure was checked by evaluating the number of iron ions per cm$^2$ in the dropcast sample. Considering the mean superficial density of the molecule[23], a sample thickness of the order of ~ 100 nm was calculated in agreement with the nominal thickness of the dropcast sample. It is worth noting that, because the sensed Mössbauer thickness is the result of the multiplication by the above-mentioned factor, only synchrotron light at grazing incidence could succeed in detecting Mössbauer spectra not only of the monolayer but also of such a thin dropcast sample.

A rather good agreement between experimental and best-fit data was found in the whole temperature range for the dropcast sample ($\chi^2$ values are comprised between 472 and 3908 for the 512 points); in spite of the worse signal-to-noise ratio, a quite good agreement was found for the monolayer sample too ($\chi^2$ values are comprised between 502 and 586 for the 512 points).

**Ab initio calculations**. The ORCA 3.0.3 program package was used to calculate the hyperfine tensors at the $^{57}$Fe nuclei. Calculations were made on one of the four crystallographically-independent molecules contained in the unit cell of the optimized structure of Fe$_4$ and on the final geometry of two out of eight AIMD trajectories (walkers 1 and 3) considered in a previous work[26]. The optimized geometries were obtained for the $M_S = 5$ Broken Symmetry state. We have limited the calculations to only two walkers because of the demanding computation time for such a large number of atoms (270). However, to avoid any loss of generality, we have chosen walkers 1 and 3 because they present very similar exchange parameters, but a different sign of the single-ion $D$ value for Fe1. The gradient-corrected (GGA) exchange-correlation functional PBE[36] was used with def2-TZVP basis set for all the elements and the RI approximation for the Coulomb operator integral evaluation employing the def2-TZVP/J auxiliary basis set[37]. It is known that the use of pure Gaussian atomic basis leads to an inaccurate descriptions of the electron density at the $^{57}$Fe nucleus[38]. For such a reason we used the core-polarized CP(PPP)[39, 40] basis function for the iron atoms. Relativistic effects were not included in the computations, since it was shown that they do not improve the quality of the computed Mössbauer parameters[40]. The integral accuracy parameter was increased to 7.0 at the Fe center in order to provide more accurate core properties. The computed isotropic hyperfine fields were scaled by a factor of 1.81 as suggested for B3LYP in [40]. In support to this choice, we have observed that an empiric factor 1.79 (hence very close to the B3LYP one) is needed to minimize the differences between the experimental and computed hyperfine fields for the central iron ion, chosen as reference. The Mulliken population analysis was performed on optimized structures obtained by AIMD (see ref. [26]) using molecular optimized basis sets (DZVP-MOLOPT-SR-GTH) along with Goedecker−Teter−Hutter pseudopotentials. The TPSS functional together with Grimme's D3 corrections was used to account for the dispersion forces. The auxiliary plane wave basis set was needed for the representation of the electronic density in the reciprocal space and the efficient solution of Poisson's equation. We truncated the plane wave basis set at 400Ry.

**Data availability**. All relevant data, including ASCII files of the recorded Mössbauer spectra, are available from the authors on request.

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

## Acknowledgements

This work was in part supported by the European Research Council through the Advanced Grant MolNanoMaS (no. 267746) and by European Union's Horizon2020 Research and Innovation Programme under grant agreement No. 737709 (FEMTO-TERABYTE, http://www.physics.gu.se/femtoterabyte). MF is grateful for financial support by ECRF (no. 2014.0708). We thank Dr. Alessandro Lunghi for fruitful discussion.

## Author contributions

A.Co., M.M., and R.S. planned the investigation. A.Co. synthesized the material while molecular films were prepared by M.M. A.Ci., A.Co., and M.M. performed the experiments with the assistance of A.Ch. and R.R. Mössbauer data were analysed by A.Ci., M.F., and G.S. with the support of A.Co. and R.S.; F.T. performed the ab initio studies. All authors contributed to the discussion and to the drafting of the manuscript.

## Additional information

**Competing interests:** The authors declare no competing financial interests.

