## [Peer Review File · Nature Communications]

Reviewers' comments:

Reviewer #1 (Remarks to the Author):

The manuscript "Synchrotron Mössbauer spectroscopy on a monolayer of single molecule magnets unveils the effects of molecule-surface interactions" by Cini et al. describes the study of tetrairon SMMs grafted on a polycrystalline gold film by synchrotron Moessbauer spectroscopy. Static and dynamic magnetic properties are extracted from extensive fitting to the experimental data.

Overall, I find the manuscript original, interesting and very well written. Most of the conclusions are supported by the experiment. However, I believe that certain statements need to be rechecked for their validity. Therefore, I recommend that the manuscript should be published in Nature Communications after the authors have made appropriate changes to it.

In more detail, I could not find any X-ray absorption data on similar samples as used in the present study. The XAS investigations in Refs 11 and 23 of the main text were performed on Au/mica or on Au(111) single crystalline substrates which provide a much better surface quality. In the present study it cannot be excluded that the distortions of the molecules are strong (and thus observable) because of a comparably rough surface. This is also consistent with the observation of randomly orientated Fe₄ molecules (last paragraph on p. 12).

Furthermore, I have quite some doubts that the information found in the deconvoluted spectra is really present in the experimental spectra, which exhibit a considerable noise due to the small amount of material. Nevertheless, it is clearly observable that in the monolayer sample the Moessbauer lines are broadened w.r.t the dropcasted sample. Couldn't this be due to a faster spin dynamics? If not, the authors should explain why. Furthermore, if the authors insist in keeping the present deconvolution, I strongly recommend that they perform a solid error and correlation analysis of the fitted peak heights and positions.

Minor points:

- Give more experimental details: What is the spot size on the sample? Further, I guess all measurements are done without magnetic field but it is not specified anywhere. Probably it does not hurt to specify the energy of the synchrotron beam as well.
- On page 6 last paragraph, the authors state that there should be 124 transitions. Please comment why this is so.
- On page 7, end of first paragraph: How is the sample texture characterized and what does it mean?
- Eq 1: I cannot recognize the magnetic dipolar interaction.
- The obtained transition rates are on the order of 100 KHz or more. I don't think that this allows to assign or compare SMM properties (of course, if one would apply a magnetic field, the situation could be different but that does not reflect the circumstances of the described experiment).
- I don't understand why there is such a strong temperature dependence of the transition rate down to 2.2 K. I would expect QTM to dominate because of the absence of magnetic field.
- I don't understand the sentence on page 15, second paragraph, "Although the signal-to-noise ratio is higher..."

Reviewer #2 (Remarks to the Author):

The current paper reports a technical tour de force – the first energy domain Mössbauer spectrum on a monolayer of molecules. As such, the technique in the future might be of broad interest to the surface science and catalysis communities.

I did have a couple technical questions that might relate to future applications:

The authors mention that deposition on thin insulating layers (e.g. MgO) rather than directly on the metal surface yields better memory effects. Yet, in this experiment they deposited directly on a gold surface. What are the technical barriers to using a thin insulator surface on top of a metal surface? Why not MgO ?

The linewidth at half maximum (FWHM) was approximately three times larger than for a radioactive source – can this be improved?

Regarding the current results:

How was the monolayer nature of the deposition confirmed?

As far as statistical analysis goes, they present very nice fits, but can they show the residuals ?

Overall, this is an excellent experimental result regarding a method that could have broad applications in the future.

Reviewer #3 (Remarks to the Author):

The authors present a combined experimental and theoretical Mossbauer study of the molecular spin dynamics in a monolayer of Fe₄ SMMs self-assembled on a polycrystalline Au surface, using an unusual experimental set up based on synchrotron gamma-rays impinging at grazing angles onto the monolayer, to enhance the sensitivity of the measure. Their results are compared with the same experiment carried out on a dropcast sample, which is shown to display the same response characteristics of a bulk sample. By presenting a detailed analysis of the experimental data based on a combination of state of the art modelling tools, including ab initio molecular dynamics to study the effects of disorder on the zero-field splitting spin Hamiltonian terms and on the hyperfine fields, and quantum master equations approaches in the linear response regime to model the fluctuating hyperfine fields at the ⁵⁷Fe sites as function of temperature, the authors convincingly show that their Mossbauer experiments can probe single-molecule spin dynamics for a monolayer of SMMs assembled on a surface to an unprecedented level of detail.

The work is carried out very competently in both its experimental and theoretical components, and the results obtained represent a significant advancement in the field of molecular magnetism probed in non-crystalline environments. I only have a few minor observations, mostly concerning the possible improvement of the clarity of presentation of the spin dynamics modelling results, and of the results of ab initio modelling of the hyperfine fields, which I believe will improve the readability of the work.

- 1) Overall, my main critique to the presentation of the theoretical analysis of the signature of the SMM spin dynamics on the Mossbauer lineshapes is that it seems to limit itself to account for the temperature evolution of the spectrum based on direct numerical simulation, but makes little efforts to explain what is the emerging microscopic picture. For instance, it would be nice a more detailed/clear discussion, in the main text of the paper, of what are the signatures in the Mossbauer spectrum of the transition from a slow-relaxation dynamics, to a fast relaxation dynamics.
- 2) Page 11, while discussing the change in lineshape of the spectrum with raising temperature, the authors claim that between 2.2K and 11K the increase in the number of lines and change in their intensity observed with increasing temperature likely reflects a spin dynamics that is dominated by the population of the excited M_S levels of the S=5 multiplet, while for higher temperatures the change in lineshape and shift towards the centre reflects a dynamics dominated by increased transition rates. This statement I believe should be further qualified and clarified, perhaps by making explicit reference to the eigenvectors and eigenvalues of the G(p)⁻¹ superoperator. As it stands, the statement is somewhat cryptic, given that fast transition rates dynamics (with respect to the observed nuclear transitions) would seem to be inextricably linked to a spin dynamics dominated by thermal population of excited states, i.e. necessary in order for well-defined/non-fluctuating populations of excited states to play a dominant role in the dynamics. In the limit of “instantaneous” transition rates, well-defined Boltzmann populations of these spin levels would be achieved. Perhaps it is just a language issue, but it would be useful if the authors could elaborate a bit more on this statement. In fact, more in general, the current theoretical analysis of the signature of the SMM spin dynamics on the Mossbauer lineshapes, as presented in the main text of the paper, seems to be limited to a statement that the temperature evolution of the spectrum is well

reproduced by direct numerical simulation using the authors' model, but little effort is made to explain what is the emerging microscopic picture. For instance, it would be nice if the author could clearly illustrate, in the main text of the paper, what are the signatures in the Mossbauer spectrum of the transition, as function of temperature, from a slow-relaxing spin dynamics, to a fast relaxing spin dynamics. Why the disappearance of the external spectral lines and shift towards the centre is a distinctive signature of the slow-to-fast relaxation spin dynamics, and it is not just a consequence of having more equalised populations of all spin levels?

- 3) Page 12, in their spin dynamics model the authors fit as a function of temperature a single transition rate between two specific spin levels ($M_S=5$ and $M_S=4$), assuming a rather simple relationship between all transition rates, i.e. assuming that all transition rates are related by a temperature-independent scaling factor (the matrix element of an appropriate spin-only operator). However, even simple phenomenological models of spin-phonon coupling rates would seem to be more complex than this. While it is true that the temperature dependence of these matrix elements is related to dynamical properties of the phonon bath only, which in turn depend on the energy structure of the phonon bath and the ensuing bath correlation functions, the transition rates depend on Fourier transforms of such bath correlation functions evaluated at the SMM energy gap associated to a given transition between the SSM M_S -levels, so that even after normalisation of a transition rate by the specific value of the transition probability (temperature independent matrix element), such rate would still have a specific temperature dependence on account of its specific energy gap. For instance, if such temperature dependence were exponential in gap/T , in the authors' approximation a logarithmic plot of different rates (associated to different gaps) as function of $1/T$ would expose a linear dependence with the same slope but different intercepts for each rate, while in reality different rates would have both different slopes (proportional to the specific gap) and different intercepts. The author should comment on this point.
- 4) Still at page 12, on a related point, I am not convinced, based on previous point 2, that the 8.9K obtained as an exponent for W_{5-4} is solely related to the bath energetic structure, while I would agree that it is not directly a measure of the spin inversion rate.
- 5) Page 13, second paragraph, the authors state that the distinctive feature of the monolayer spectrum of displaying approximately the same linewidth, larger than in the dropcast sample, can only be explained in terms of an inhomogeneous distribution of electric parameters, since a distribution of hyperfine fields would affect the external lines six times more than the internal ones. This sounds like a simple statement to explain, so the authors should elaborate on this, and briefly explain why that is the case.
- 6) Page 14, 15 and Ab Initio Methods section at page 18: the authors are performing open-shell ab initio DFT calculations of the hyperfine fields at the ^{57}Fe nuclei for a monolayer of Fe_4 SMMs, in order to further unravel the influence of structural distortions in a non-crystalline environment, and also to probe the electronic effects produced by the Au surface. As found by the authors, the hyperfine field is dominated by the Fermi Contact (FC) contribution, which however can only probe the spin density induced at the nuclear position, i.e. the spin-polarisation of s atomic orbitals of the ^{57}Fe atoms. Given the size of

this molecule, especially if Au surface is taken into account, multiconfigurational wavefunction approaches, which would be the rigorous ab initio approach for a system (i) which has an antiferromagnetic (hence intrinsically multiconfigurational) ground state (ii) for which the authors are interested in reproducing tiny spin-polarization effects of core s-electrons, are clearly out of reach, and it is necessary to make use of approximate DFT methodologies, which have the drawback that in the Kohn-Sham approximation are essentially one-determinant approaches. Hence, the only way to reproduce the crucial physical properties the authors are interested in (antiferromagnetically coupled spin states, and core-electron spin-polarization effects), is to break spin-symmetry in the DFT calculation, a broken-symmetry (BS) DFT approach, which needs to be optimised for the particular MS spin configuration one wishes to model. Assuming the calculations presented here were in fact BS-DFT calculations, this should be clearly stated by authors (I could not quite find reference to this in the paper). Also, more details as to how the $M_S=5$ ground state configuration was optimised by the authors, via spin-orbital localisation or else, should be provided.

- 7) Still related to the DFT calculations of hyperfine coupling fields: while for the isolated (but distorted) molecules the authors are using a broken symmetry B3LYP functional with core-enriched basis functions to more accurately reproduce core spin polarization effects, the periodic boundary condition DFT calculations describing the effect of the Au surface were carried out via pseudo-potential Gaussian Plane Waves GPW-DFT methodologies, for which there do not seem to be many details reported in the paper or in the supplementary information file. For instance, a natural question is: if the authors are using pseudo-potentials, which typically replace the complicated structure of the wavefunction characterising core electrons with simpler mathematical functions that only match the atomic wavefunction beyond a given threshold radius, that would seem a less than ideal way to achieve a satisfactory, even qualitatively, description of FC-dominated hyperfine fields. So one wonders if the reduced value for hyperfine fields obtained by the authors is actually due to their inclusion of the Au surface, or whether it is a consequence of the less than ideal representation of the core electron properties provided by pseudo-potentials. The authors should (i) give more details about their calculations (ii) comment on this possible shortcoming of the approach.

Authors rebuttal letter for manuscript NCOMMS-17-19771-T entitled “Synchrotron Mössbauer spectroscopy on a monolayer of single molecule magnets unveils the effects of molecule-surface interactions”

We thank reviewers for the appreciation of our work and for the punctual and constructive comments which have helped us to significantly improve the clarity of our manuscript.

The original comments are reported in blue, our answers in black and the new text (when not too long) in red.

Reviewer #1

1) In more detail, I could not find any X-ray absorption data on similar samples as used in the present study. The XAS investigations in Refs 11 and 23 of the main text were performed on Au/mica or on Au(111) single crystalline substrates which provide a much better surface quality. In the present study it cannot be excluded that the distortions of the molecules are strong (and thus observable) because of a comparably rough surface. This is also consistent with the observation of randomly orientated Fe₄ molecules (last paragraph on p. 12).

The reviewer is in principle right. The requirement of a flat and rigid surface to perform experiments at very grazing incidence have not left us a great choice and a parallel XAS/XMCD investigation was not possible on the same samples and has not been performed in previous or successive experiments. The use of large scale facility resources requires a careful balance between costs and relevance of additional information acquired through the experiment. We feel that we can safely exclude that XAS/XMCD could have revealed additional features due to the roughness of the substrate, beyond those related to preferential orientation, given that XAS/XMCD spectra superimposable to those of the bulk have also been obtained when grafting Fe₄ to Au nanoparticles (see ref xx), which could actually represent the opposite extreme compared to flat (111) terraces for which similar results have already been reported. We also agree that we cannot exclude that the roughness of the – necessarily - employed substrate plays an important role. Nevertheless the ab initio molecular dynamics simulations, performed on a flat Au(111) surface suggest that distortions are also present on a flat surface.

To clarify this point a sentence has been added in the Materials and Methods section.

“ No XAS/XMCD data are available for a monolayer assembled on the substrates employed here. However, we recall that Fe₄ molecules chemisorbed on well reconstructed Au(111) surfaces or tethered to Au nanoparticles³⁴ both give XAS/XMCD spectra that do not differ appreciably from those recorded in the bulk phase. (Ref. 34 doi: 10.1002/sml.201301617”

2) Furthermore, I have quite some doubts that the information found in the deconvoluted spectra is really present in the experimental spectra, which exhibit a considerable noise due to the small amount of material. Nevertheless, it is clearly observable that in the monolayer sample the Moessbauer lines are broadened w.r.t the dropcasted sample. Couldn't this be due to a faster spin dynamics? If not, the authors should explain why. Furthermore, if the authors insist in keeping the present deconvolution, I strongly recommend that they perform a solid error and correlation analysis of the fitted peak heights and positions.

The discussion of the spectral features due to spin dynamics has been significantly extended in Supplementary Note 2 and 4 and with addition of Supplementary Fig. 8 and 14. Moreover, an analysis

performed without introducing a distribution of the electric parameters determines an anomalous behavior of the transition rates with the temperature (the transition rates increased upon lowering T) therefore pointing to an unrealistic situation.

3) Give more experimental details: What is the spot size on the sample? Further, I guess all measurements are done without magnetic field but it is not specified anywhere. Probably it does not hurt to specify the energy of the synchrotron beam as well.

These data were already present in the Supplementary Information but they have been partially rewritten and included in the method section.

4) On page 6 last paragraph, the authors state that there should be 124 transitions. Please comment why this is so.

While more details on the size of the difference matrices are provided in Supplementary Notes, in the main text this point has been clarified by adding at page 6:

Given that the Fe₄ molecule has total spin state $S = 5$, we expect each iron atom to contribute to the spectra with six hyperfine-split lines per $|\pm M_S\rangle$ doublet ($M_S = 1, 2, \dots, 5$) plus two quadrupole-split lines arising from the $|M_S = 0\rangle$ state, thus yielding $4(6 \cdot 5 + 2) = 128$ lines.

5) On page 7, end of first paragraph: How is the sample texture characterized and what does it mean?

This point has been clarified in the text by adding the following sentence:

'The latter is a parameter quantifying an eventual macroscopic orientation of the molecules on the surface. In particular, the possibility to sense a texture, which has a signature in the relative intensities of the lines, is based on the orientation of the quantization axis of the molecule with respect to the γ -ray polarization.'

6) Eq 1: I cannot recognize the magnetic dipolar interaction.

The second term is actually containing all anisotropic contribution to intramolecular interactions. In Fe₄ where the single ion have high spins d⁵ configuration deviations from $g=2$ are negligible and the largely dominating contribution to this term is the dipolar one.

A sentence has been added after eq. 1 to clarify this point:

“The first term represents the isotropic exchange interaction, the second one contains the anisotropic part of intramolecular interactions, which are mainly of dipolar origin in assemblies of high spin $3d^5$ ions, while ... “

7) The obtained transition rates are on the order of 100 kHz or more. I don't think that this allows to assign or compare SMM properties (of course, if one would apply a magnetic field, the situation could be different but that does not reflect the circumstances of the described experiment).

The entire discussion on spin dynamics effects on the spectra has been rewritten and expanded, Specifically for the point mentioned here we can say that, in general, the $|ms=\pm 5\rangle \rightarrow |ms=\pm 4\rangle$ transition investigated here is only a step in the process of magnetization reversal monitored by

magnetometry and thus cannot be directly compared with relaxation times extracted from ac susceptibility, as clearly stated in the main text. However, it must be noted that the application of a static field is not making substantial differences. In fact, ac susceptibility up to 10 kHz has been regularly used to characterize the dynamics of Fe₄ systems showing that the dynamics in zero field follows the Arrhenius law with an effective barrier that well compare with the theoretical one expected from the magnetic anisotropy ($|D|S^2$). The effect of the field is not relevant at the investigated temperatures.

8) I don't understand why there is such a strong temperature dependence of the transition rate down to 2.2 K. I would expect QTM to dominate because of the absence of magnetic field.

As also mentioned above, in Fe₄ systems, widely investigated in the crystalline phase by microSQUID techniques down to very low temperature, even in zero field the pure tunneling regime is only reached below 400 mK. At 2.2 K ac susceptibility data indicate an Arrhenius behavior with an effective barrier in zero field only slightly reduced compared to the theoretical value $|D|S^2$.

9) I don't understand the sentence on page 15, second paragraph, "Although the signal-to-noise ratio is higher..."

We apologize for the mistake. We have corrected the sentence as

“Despite the much weaker signal-to-noise ratio, it is evident that, on increasing the temperature, the external lines lose intensity faster than in the dropcast sample. This suggests that the $|D|$ value is reduced upon grafting, as directly confirmed by fitting the Mössbauer spectrum recorded at the lowest temperature ($D = -0.49(6)$ K, with the E/D ratio held fixed at the value found in the dropcast sample to avoid overparametrization in the fitting of poorly resolved spectra).”

Reviewer #2:

1) The authors mention that deposition on thin insulating layers (e.g. MgO) rather than directly on the metal surface yields better memory effects. Yet, in this experiment they deposited directly on a gold surface. What are the technical barriers to using a thin insulator surface on top of a metal surface? Why not MgO ?

In the introduction, the possibility to modulate spin dynamics by the substrate was mentioned to provide to the general reader the context and relevance of investigation of monolayer of magnetic molecules on surfaces. The suggestion of the reviewer is very interesting but, at the moment, not feasible. The beamline is not equipped with an in-situ evaporation chamber to prepare the substrate and evaporate a monolayer of molecules. We have been therefore obliged to adopt the self-assembly procedure from solution and we chose the previously investigated protected thiols – gold pair.

2) The linewidth at half maximum (FWHM) was approximately three times larger than for a radioactive source – can this be improved?

The linewidth of the Synchrotron Moessbauer Source can be adjusted in a compromise with the beam intensity [ref. 28]. Typically, an improvement of the linewidth by a half of the natural linewidth decreases the count rate by a factor of two. For current experiment, where the extreme resolution is not an issue,

we have chosen the given value of the linewidth for the benefit of a shorter acquisition time. In more resolution-demanding studies, the linewidth can be improved down to about 1.5 natural linewidths.

In the Materials and Methods section we have added the sentence:

For the present study an acceptable compromise between linewidth and acquisition time was achieved by setting the linewidth at half maximum (FWHM) at a value approximately three times larger than for a radioactive source. In these conditions, an intensity of about 1.5×10^4 photons per second was obtained on the 14.4 keV γ -line.”

3) Regarding the current results: How was the monolayer nature of the deposition confirmed?

The monolayer sample used for the SMS experiments has not been subjected to any other characterization. Our extensive experience on Self Assembled Monolayers (SAM) of these type of molecules has allowed us to develop a reliable deposition protocol which, however, requires that the monolayer is prepared on site from a fresh solution and transferred with a portable glove box (under Ar atmosphere) to the sample holder and quickly inserted in the cryostat.

The most suitable techniques to assess the formation of a monolayer, commonly used by us, are ToF SIMS and XPS. Also in this case freshly prepared SAMs are necessary. Unfortunately, these two techniques are not available at the beamline.

An ex-post characterization, i.e. recovering the sample after several days of measurements, was also not possible because the cryostat does not allow to transfer back the sample in a glove box without the cold sample-holder being exposed to air/humidity.

4) As far as statistical analysis goes, they present very nice fits, but can they show the residuals ?

The residuals have been reported in the Supplementary Figures 5, 6, 9 and 10 for selected cases.

5) Overall, this is an excellent experimental result regarding a method that could have broad applications in the future.

We thank again the referee for the words of appreciation for our work.

Reviewer #3

1) Overall, my main critique to the presentation of the theoretical analysis of the signature of the SMM spin dynamics on the Mossbauer lineshapes is that it seems to limit itself to account for the temperature evolution of the spectrum based on direct numerical simulation, but makes little efforts to explain what is the emerging microscopic picture.

For instance, it would be nice a more detailed/clear discussion, in the main text of the paper, of what are the signatures in the Mossbauer spectrum of the transition from a slow-relaxation dynamics, to a fast relaxation dynamics.

Once more, we thank the reviewer for having stimulated a deeper discussion on the dynamic properties of the system. These points have been more extensively developed.

The points raised by the reviewer have been carefully considered and Supplementary Notes 2 and 4 hopefully clarifying the discussion. Calculated spectra assuming different scenarios are reported in Supplementary Figure 8 and a simple model to show the three possible dynamic regimes has been introduced with the results shown in Supplementary Figure 14. In the main text, this aspect has been mainly improved by rewriting pages 12 and 13, as well in other specific points.

2) Page 11, while discussing the change in lineshape of the spectrum with raising temperature, the authors claim that between 2.2K and 11K the increase in the number of lines and change in their intensity observed with increasing temperature likely reflects a spin dynamics that is dominated by the population of the excited MS levels of the S=5 multiplet, while for higher temperatures the change in lineshape and shift towards the centre reflects a dynamics dominated by increased transition rates. This statement I believe should be further qualified and clarified, perhaps by making explicit reference to the eigenvectors and eigenvalues of the $G(p)$ -1 superoperator. As it stands, the statement is somewhat cryptic, given that fast transition rates dynamics (with respect to the observed nuclear transitions) would seem to be inextricably linked to a spin dynamics dominated by thermal population of excited states, i.e. necessary in order for well defined/non-fluctuating populations of excited states to play a dominant role in the dynamics. In the limit of “instantaneous” transition rates, well-defined Boltzmann populations of these spin levels would be achieved. Perhaps it is just a language issue, but it would be useful if the authors could elaborate a bit more on this statement. In fact, more in general, the current theoretical analysis of the signature of the SMM spin dynamics on the Mossbauer lineshapes, as presented in the main text of the paper, seems to be limited to a statement that the temperature evolution of the spectrum is well reproduced by direct numerical simulation using the authors’ model, but little effort is made to explain what is the emerging microscopic picture. For instance, it would be nice if the author could clearly illustrate, in the main text of the paper, what are the signatures in the Mossbauer spectrum of the transition, as function of temperature, from a slow relaxing spin dynamics, to a fast relaxing spin dynamics. Why the disappearance of the external spectral lines and shift towards the centre is a distinctive signature of the slow to-fast relaxation spin dynamics, and it is not just a consequence of having more equalised populations of all spin levels?

This point is related to the previous one and has been fully addressed by expanding the Supplementary Information and modifying the main text as explained in the previous answer.

3) Page 12, in their spin dynamics model the authors fit as a function of temperature a single transition rate between two specific spin levels (MS=5 and MS=4), assuming a rather simple relationship between all transition rates, i.e. assuming that all transition rates are related by a temperature-independent scaling factor (the matrix element of an appropriate spin-only operator). However, even simple phenomenological models of spin-phonon coupling rates would seem to be more complex than this. While it is true that the temperature dependence of these matrix elements is related to dynamical properties of the phonon bath only, which in turn depend on the energy structure of the phonon bath and the ensuing bath correlation functions, the transition rates depend on

Fourier transforms of such bath correlation functions evaluated at the SMM energy gap associated to a given transition between the SSM MS-levels, so that even after normalisation of a transition rate by the specific value of the transition probability (temperature independent matrix element), such rate would still have a specific temperature dependence on account of its specific energy gap. For instance, if such temperature dependence were exponential in gap/T , in the authors' approximation a logarithmic plot of different rates (associated to different gaps) as function of $1/T$ would expose a linear dependence with the same slope but different intercepts for each rate, while in reality different rates would have both different slopes (proportional to the specific gap) and different intercepts. The author should comment on this point.

This point has been extensively clarified in the Supplementary Information by reporting the specific dependences of the transition rates for direct and indirect processes on the temperature, as well as on the electronic energy gap. In the main text these dependences are recalled and discussed in relation to the trends experimentally found, see pages 12 and 13 of the main text. As an example we paste here the brief explanation of the employed approximation, given at page 12 of the main text:

“In particular, the parameters determining the thermal dependence of the spectra are proportional to the Fourier transform of the correlation function of F_b evaluated at the transition frequencies. This quantity can be evaluated at zero frequency in the usual approximation where the linewidth of bath states is larger than the energy separation between the electronic spin states. More details are provided in Supplementary Note 4, where both direct (single-phonon) and indirect (multi-phonon) processes are considered. “

4) Still at page 12, on a related point, I am not convinced, based on previous point 2, that the 8.9K obtained as an exponent for W5-4 is solely related to the bath energetic structure, while I would agree that it is not directly a measure of the spin inversion rate.

This point has been clarified in the Supplementary Information and in the main text pages 12 and 13. In particular, to be more accurate, the specific dependence of the transition rates on the temperature for direct and indirect processes in the limit of low temperature has been considered, instead of the simple exponential (see new equations 4 and 5).

5) Page 13, second paragraph, the authors state that the distinctive feature of the monolayer spectrum of displaying approximately the same linewidth, larger than in the dropcast sample, can only be explained in terms of an inhomogeneous distribution of electric parameters, since a distribution of hyperfine fields would affect the external lines six times more than the internal ones. This sounds like a simple statement to explain, so the authors should elaborate on this, and briefly explain why that is the case.

This point has been clarified in the SI and in the main text. The factor close to six arises from the different nuclear g factor of the ground and excited state. The Supplementary Note 2 now reads:

In particular, for the sixth line the energy value deriving from the separation between the levels splitted only by the magnetic interaction is proportional to

$$\left(\frac{3}{2}g_{ex} - \frac{1}{2}g_g\right)B_{hf}$$

where B_{hf} is the hyperfine magnetic field and g_{ex} and g_g are the nuclear g values of the excited and ground states, respectively. For the fourth line the relation is instead:

$$\left(-\frac{1}{2}g_{ex} - \frac{1}{2}g_g\right)B_{hf}$$

For ^{57}Fe the values of g_{ex} and g_g are -0.1031 and 0.1808, respectively. Therefore, the ratio between the sixth and fourth lines is close to six.

6) Page 14, 15 and Ab Initio Methods section at page 18: the authors are performing open-shell ab initio DFT calculations of the hyperfine fields at the ^{57}Fe nuclei for a monolayer of Fe4 SMMs, in order to further unravel the influence of structural distortions in a non-crystalline environment, and also to probe the electronic effects produced by the Au surface. As found by the authors, the hyperfine field is dominated by the Fermi Contact (FC) contribution, which however can only probe the spin density induced at the nuclear position, i.e. the spin-polarisation of s atomic orbitals of the ^{57}Fe atoms. Given the size of this molecule, especially if Au surface is taken into account, multiconfigurational wave functions approaches, which would be the rigorous ab initio approach for a system (i) which has an antiferromagnetic (hence intrinsically multiconfigurational) ground state (ii) for which the authors are interested in reproducing tiny spin-polarization effects of core s-electrons, are clearly out of reach, and it is necessary to make use of approximate DFT methodologies, which have the drawback that in the Kohn-Sham approximation are essentially one-determinant approaches. Hence, the only way to reproduce the crucial physical properties the authors are interested in (antiferromagnetically coupled spin states, and core-electron spin-polarization effects), is to break spin-symmetry in the DFT calculation, a broken-symmetry (BS) DFT approach, which needs to be optimised for the particular MS spin configuration one wishes to model. Assuming the calculations presented here were in fact BS-DFT calculations, this should be clearly stated by authors (I could not quite find reference to this in the paper). Also, more details as to how the MS=5 ground state configuration was optimised by the authors, via spin-orbital localisation or else, should be provided.

We thank the referee for his relevant analysis and hints to make this part clearer.

The AIMD calculations performed in our previous paper (J. Mater. Chem. C, 2014, 2, 8333-8343) were all performed on a $M_S = 5$ Broken Symmetry through an initial spin-orbital localization guess. In order to make this information readily available to the reader we have added the following sentence in the computational section:

“.. considered in ref.25 The optimized geometries have been obtained for the $M_S = 5$ Broken Symmetry state. ...”

7) Still related to the DFT calculations of hyperfine coupling fields: while for the isolated (but distorted) molecules the authors are using a broken symmetry B3LYP functional with core-enriched basis functions to more accurately reproduce core spin polarization effects, the periodic boundary condition DFT calculations describing the effect of the Au surface were carried out via pseudo-potential Gaussian Plane Waves GPW-DFT methodologies, for which there do not seem to be many details reported in the paper or in the supplementary information file. For instance, a natural question is: if the authors are using pseudo-potentials, which typically replace the complicated structure of the wavefunction characterising core electrons with simpler mathematical functions that only match the atomic wavefunction beyond a

given threshold radius, that would seem a less than ideal way to achieve an satisfactory, even qualitatively, description of FC-dominated hyperfine fields. So one wonders if the reduced value for hyperfine fields obtained by the authors is actually due to their inclusion of the Au surface, or whether it is a consequence of the less than ideal representation of the core electron properties provided by pseudo-potentials. The authors should (i) give more details about their calculations (ii) comment on this possible shortcoming of the approach.

The pseudo-potential/GPW-DFT results, indeed, indicate that a small but sizable reduction of the spin density is observed when the electronic effect of the surface is considered. No direct information on core s orbitals can be, obviously, achievable within such a computational approach. However, an indirect quantification of the reduction of the polarization of the core s orbitals can be, therefore, estimated by the reduction of the computed spin densities in the pseudo-potential/GPW-DFT calculations. Moreover, to avoid any spurious effect due to more approximated method the electronic effect of the substrate (see last row in Table 1) has been computed comparing both $Fe_4@Au$ and $Fe_4@Au$ at the same level of approximation.

To give more details about such computational protocol, we have added in “Materials and Methods” section the following text:

“...The Mulliken population analysis was performed on optimized structures obtained by AIMD (see ref. 26 and references therein) using molecular optimized basis sets (DZVP-MOLOPT-SR-GTH) along with Goedecker–Teter–Hutter pseudopotentials. The TPSS functional together with Grimme’s D3 corrections was used to account for the dispersion forces. The auxiliary plane wave basis set was needed for the representation of the electronic density in the reciprocal space and the efficient solution of Poisson’s equation. We truncated the plane wave basis set at 400 Ry. “.

In the main text, the following statement has been added to the paper:

pag 14: “obtained, at the same level of calculation to avoid spurious effect due to the more approximated method, ..”

pag 14: “...They indicate that a small but sizable reduction of the spin density occurs due to a purely electronic effect of the gold substrate supporting a decrease of the polarization of the core s orbitals of the iron ions. Such a reduction of the hyperfine...”

Reviewer #1 (Remarks to the Author):

The authors have amended their manuscript and answered satisfactorily to nearly all of my criticism. However, I still disagree regarding my point 7. In my opinion, the last paragraph of the introduction sounds like synchrotron Moessbauer spectroscopy revealed slow relaxation of magnetization. I believe this is at least misleading, or even wrong, since it uses a term which is commonly employed for the relaxation between the spin-up and spin-down ground states of an SMM.

Therefore I can only recommend the manuscript for publication after both of the following two points have been resolved:

1) The authors remove or correct the first sentence of the last paragraph of the introduction ("..., which clearly revealed slow magnetic relaxation effects."). If the authors insist in mentioning the term slow magnetic relaxation they should specify the states involved in the observed relaxation process and also the timescale which is obtained from the present Moessbauer experiment and not from other experiments.

2) The authors revise the last sentence ("...the molecule retains "... and slow relaxation of the magnetization on the surface...") in the same way to avoid any misunderstandings.

Reviewer #2 (Remarks to the Author):

The authors have clearly spent considerable time responding to the points raised by the different referees. It is unfortunate that additional sample characterization such as XAS or XPS was not available. However, it is still a valuable contribution to illustrate the potential of surface sensitive Mössbauer spectroscopy.

I recommend publication once final issues raised by responses from other referees are dealt with.

Reviewer #3 (Remarks to the Author):

In their revised manuscript, the authors have satisfactorily addressed the questions previously raised, hence I can recommend the publication of this nice work as is.

Rebuttal letter of manuscript NCOMMS-17-19771A

Reviewer #1 (Remarks to the Author):

The authors have amended their manuscript and answered satisfactorily to nearly all of my criticism. However, I still disagree regarding my point 7. In my opinion, the last paragraph of the introduction sounds like synchrotron Moessbauer spectroscopy revealed slow relaxation of magnetization. I believe this is at least misleading, or even wrong, since it uses a term which is commonly employed for the relaxation between the spin-up and spin-down ground states of an SMM.

Therefore I can only recommend the manuscript for publication after both of the following two points have been resolved:

- 1) The authors remove or correct the first sentence of the last paragraph of the introduction ("..., which clearly revealed slow magnetic relaxation effects."). If the authors insist in mentioning the term slow magnetic relaxation they should specify the states involved in the observed relaxation process and also the timescale which is obtained from the present Moessbauer experiment and not from other experiments.
- 2) The authors revise the last sentence ("...the molecule retains "... and slow relaxation of the magnetization on the surface...") in the same way to avoid any misunderstandings.

Our response

Concerning the only comment of reviewer #1 we agree that a local spectroscopic probe like Mössbauer – or paramagnetic resonances, muons rotation spectroscopy etc. - cannot provide direct information on a thermodynamic property like magnetization. The time-window is also dictated by the technique and cannot be varied at will as in magnetometry techniques.

Having said that, we would like to remind here that Mössbauer spectroscopy is a well-established tool in molecular magnetism. As early as in 1996 it was employed to identify the second discovered Single-Molecule Magnets, Fe₈. The latter, unfortunately not processable in monolayers, has been one of the most investigated SMM (10.1126/science.284.5411.133, 10.1038/nature10314, to mention a few). Mössbauer spectroscopy is also used to investigate blocking of the magnetization in nanoparticles such as ferritin.

We believe that the sensitivity of Mossbauer spectroscopy to both intra-well and inter-well transitions – in the double well energy potential created by magnetic anisotropy – is now clear thanks to the material we have added in the previous revision (see Supplementary Figure 15).

To avoid any further misunderstanding the sentence

"..., which clearly revealed slow magnetic relaxation effects."

has been replaced by:

"...spectroscopy, which revealed magnetic features in zero field typical of SMM behaviour."

A call to the specific time-window of Mössbauer already appears a few line above and is not needed here.

At the end of the discussion the sentence

"...the molecule retains "... and slow relaxation of the magnetization on the surface..."

has been replaced by:

"...the molecule retains on the surface an $S = 5$ ground state and slow spin dynamics comparable to that of the bulk phase, thus justifying the magnetic robustness of this class of SMMs."